# How large is the universe of RNA-like motifs? A clustering analysis of RNA graph motifs using topological descriptors

Rui Wang [1], Tamar Schlick [1,2,3,4]*

**1** Simons Center for Computational Physical Chemistry, New York University, New York, New York, United States of America, **2** Department of Chemistry, New York University, New York, New York, United States of America, **3** Courant Institute of Mathematical Sciences, New York University, New York, New York, United States of America, **4** New York University-East China Normal University Center for Computational Chemistry, New York University Shanghai, Shanghai, China

* schlick@nyu.edu

**Data availability statement:** The code and data for the feature and clustering algorithms are

## Abstract

Identifying novel and functional RNA structures remains a significant challenge in RNA motif design and is crucial for developing RNA-based therapeutics. Here we introduce a computational topology-based approach with unsupervised machine-learning algorithms to estimate the database size and content of RNA-like graph topologies. Specifically, we apply graph theory enumeration to generate all 110,667 possible 2D dual graphs for vertex numbers ranging from 2 to 9. Among them, only 0.11% (121 dual graphs) correspond to approximately 200,000 known RNA atomic fragments/substructures (collected in 2021) using the RNA-as-Graphs (RAG) framework. The remaining 99.89% of the dual graphs may be RNA-like or non-RNA-like. To determine which dual graphs in the 99.89% hypothetical set are more likely to be associated with RNA structures, we apply computational topology descriptors using the Persistent Spectral Graphs (PSG) method to characterize each graph using 19 PSG-based features and use clustering algorithms that partition all possible dual graphs into two clusters. The cluster with the higher percentage of known dual graphs for RNA is defined as the "RNA-like" cluster, while the other is considered as "non-RNA-like". The distance between each dual graph and the center of the RNA-like cluster represents the likelihood of it belonging to RNA structures. From validation, our PSG-based RNA-like cluster includes 97.3% of the 121 known RNA dual graphs, suggesting good performance. Furthermore, 46.017% of the hypothetical RNAs are predicted to be RNA-like. Among the top 15 graphs identified as high-likelihood candidates for novel RNA motifs, 4 were confirmed from the RNA dataset collected in 2022. Significantly, we observe that all the top 15 RNA-like dual graphs can be separated into multiple subgraphs, whereas the top 15 non-RNA-like dual graphs tend not to have any subgraphs (subgraphs preserve pseudoknots and junctions). Moreover, a significant topological difference between top RNA-like and non-RNA-like graphs is evident when comparing their topological features (e.g., Betti-0 and Betti-1 numbers). These findings provide valuable insights into the size of the RNA motif universe and RNA design

available at the public repository https://github.com/wangru25/PSGRNAClustering. The RNA inverse folding using dual graph representations package is available at https://github.com/Schlicklab/Dual-RAG-IF/tree/main.

**Funding:** Support from the National Institutes of Health, National Institute of General Medical Sciences Award R35-GM122562, National Science Foundation Awards (DMS-215177 and DMS-2330628) from the Division of Mathematical Sciences, and Philip-Morris USA Inc to T.S. is gratefully acknowledged. The funders had no role in study design, data collection and analysis, decision to publish, or preparation of the manuscript. R.W. is grateful for the support from the Simons Foundation and the Simons Center for Computational Physical Chemistry (SCCPC) at New York University.

**Competing interests:** The authors have declared that no competing interests exist.

strategies, offering a novel framework for predicting RNA graph topologies and guiding the discovery of novel RNA motifs, perhaps anti-viral therapeutics by subgraph assembly.

## Author summary

This work tackles a key question in RNA motif design: how large is the universe of RNA-like structures? To explore this, we develop a computational framework that uses graph theory and topological data analysis to estimate the size and content of the universe of RNA-like graph topologies. Specifically, we generate 110,667 possible 2D dual graph topologies. Among these, only 0.11% dual graphs correspond to approximately 200,000 known RNA atomic fragments/substructures. To evaluate the remaining 99.89% of the dual graphs that may or may not correspond to RNA structures, we use persistent spectral graph features and machine learning to partition all possible dual graphs into "RNA-like" and "non-RNA-like" clusters. Our method accurately identifies 97.3% of known RNA structures as RNA-like and predicts that 46.017% of the hypothetical RNAs could be potential RNA motifs. We also identify 15 high-likelihood candidates for novel RNA structures, four of which were confirmed in newly collected data in 2022. Importantly, we discover that all top RNA-like graphs tend to break down into smaller functional substructures that preserve pseudoknots and junctions. This framework opens new directions for rational RNA design and the discovery of RNA-based therapeutics.

## Introduction

Ribonucleic acid (RNA) molecules are essential biomolecules that play critical roles in various biological processes, including protein synthesis, gene expression, gene regulation, RNA editing, and RNA interference [1]. This functional versatility is largely due to RNA's ability to fold into complex secondary and tertiary structures, which allows RNAs to perform diverse functions within cells [2]. Consequently, designing novel RNA secondary or tertiary structures capable of executing specific functions has become a major focus of research. Traditional RNA structure prediction methods primarily rely on thermodynamic models that minimize free energy to predict the most stable RNA conformations [3,4]. While these methods are effective, recent advances in deep learning techniques, such as convolutional neural networks (CNNs) and recurrent neural networks (RNNs), have further accelerated the development of RNA structure prediction tools [5,6]. These deep learning models leverage large datasets to learn complex folding patterns, offering improved accuracy over traditional methods. However, while these computational approaches excel at predicting protein structures, they are limited to RNA structure prediction due to data paucity and fall short in providing a comprehensive understanding of the overall diversity and topological features of RNA molecules. Thus, there is a growing need for a more holistic approach to RNA structure analysis and design – one that systematically explores and classifies a wide range of possible RNA topologies, and that is easily scalable to the increasing amount of biological data.

To address this need, we applied our previously developed coarse-grained framework, RNA-as-Graphs (RAG), to map existing RNA atomic fragments and substructures to 2D dual graphs with vertices ranging from 2 to 9 [7–9]. In this framework, double-stranded helical

stems are represented as vertices, while single-stranded regions that connect secondary elements, such as bulges, loops, and junctions, are represented as edges. This approach provides a concise coarse-grained representations of RNA structures as dual graphs, capturing key structural features while simplifying the complexities inherent in traditional secondary and tertiary representations. The existing RNA dual graphs generated through this method are part of a larger dataset encompassing all 110,667 possible dual graphs with vertices in the same size range. This extensive dataset raises intriguing questions: 1) For those dual graphs that do not correspond to any known RNA structures (i.e., hypothetical dual graphs), can we effectively determine which are more RNA-like versus non-RNA-like? 2) What is the size of the possible RNA motif space?

To explore these questions, we initially calculated graph features and applied unsupervised clustering methods, including Partitioning Around Medoids (PAM) and $k$-means, to categorize hypothetical dual graphs into RNA-like and non-RNA-like groups [8]. This approach successfully classified 72–77% of all existing graph topologies. One year later, we enhanced the accuracy of classifying RNA-like graphs by incorporating updated features derived from Fiedler vectors and a new scoring model [10]. While these graph-based features offer a simplified and structured way to explore RNA motifs, they may miss subtle topological and geometric nuances that are crucial for RNA functionality.

To capture these nuances, we employed a computational topology method called Persistent Spectral Graph (PSG) [11,12] to extract comprehensive topology-based and geometry-based features of dual graphs. PSG has proven effective in capturing meaningful structural information of biomolecules, including proteins and RNAs [13–15]. Specifically, PSG involves creating a filtration of simplicial complex $K$ and building a series of $q$-order persistent Laplacian on $K$. The number of 0-eigenvalues of the $q$th-order persistent Laplacian indicates the number of $q$-cocycles in a given point-cloud dataset. For instance, the number of 0, 1, and 2-cocycles respectively corresponds to the number of connected components, cycles, and holes in 3D. Additionally, the non-zero eigenvalues of the $q$th-order persistent Laplacian will further reveal geometric shape evolution information about the data. Details are provided in S1.1 and S1.2 of S1 Methods.

Building on the PSG framework, we have developed a clustering approach using 19 PSG-based features to partition the graphs into RNA-like and non-RNA-like clusters. Our approach significantly outperforms previous methods by achieving higher accuracy in clustering known RNA graphs and identifying high-likelihood candidates for novel RNA motifs. Notably, our results show that the RNA-like cluster generated by our model contains 97.3% of known RNA dual graphs compared to 88.3% in our earlier work [10] (Note: This comparison is based on dual graph datasets where the number of vertices ranges from 4 to 9. This range is chosen because PSG-based methods require a minimum of 4 vertices for generation. The accuracy reported in [10] is 97.06% on a dataset with vertices ranging from 2 to 9). Among the top 15 graphs identified as high-likelihood candidates for novel RNA motifs, four were quickly validated against a new 3D RNA dataset collected in 2022 with 181 dual graphs [16]. Clustering also suggests that the size of the RNA-like motif universe is no less than 46%.

Furthermore, the distinct topological differences observed between the partitionable RNA-like and few (irreducible) non-RNA-like graphs offer valuable insights for future RNA motif design. These findings underscore the potential of our method to guide RNA-based therapeutic development and synthetic biology by facilitating the discovery and engineering of novel RNA structures. The intriguing result that about half hypothetical RNA motifs of the motif atlas are RNA-like underscores the modularity of RNA molecules (since they are built from subgraph motifs) and their hierarchical nature.

## Methods

### Dual graph representation of RNA structures

In 2003, we developed the "RNA-As-Graphs" (RAG) framework [17] to map the existing 3D structures of RNA molecules to their corresponding 2D dual graph representations (see Fig 1). This mapping process is both non-injective and non-surjective, meaning that multiple distinct 2D structures can correspond to the same 2D dual graph (non-injective), and some 2D dual graph topologies may not correspond to existing 3D RNA structures (non-surjective). This dual graph representation of RNA structures helps reconcile three topological RNA modules: tree (edge-cut with two edges), pseudoknot (edge-cut with three edges), and bridge (one-edge-connected). Our tree library is more intuitive but cannot represent pseudoknots. Essentially, our dual graphs are a coarse-graining approach for representing RNA motifs by following the "planar dual graph rules":

1. Represent a double-stranded helical stem as a vertex,
2. Represent a single strand that may occur in segments connecting the secondary elements (e.g., bulges, loops, junctions, and stems) as an edge,
3. No representation is required on the 3 and 5 ends of the RNA secondary structure.

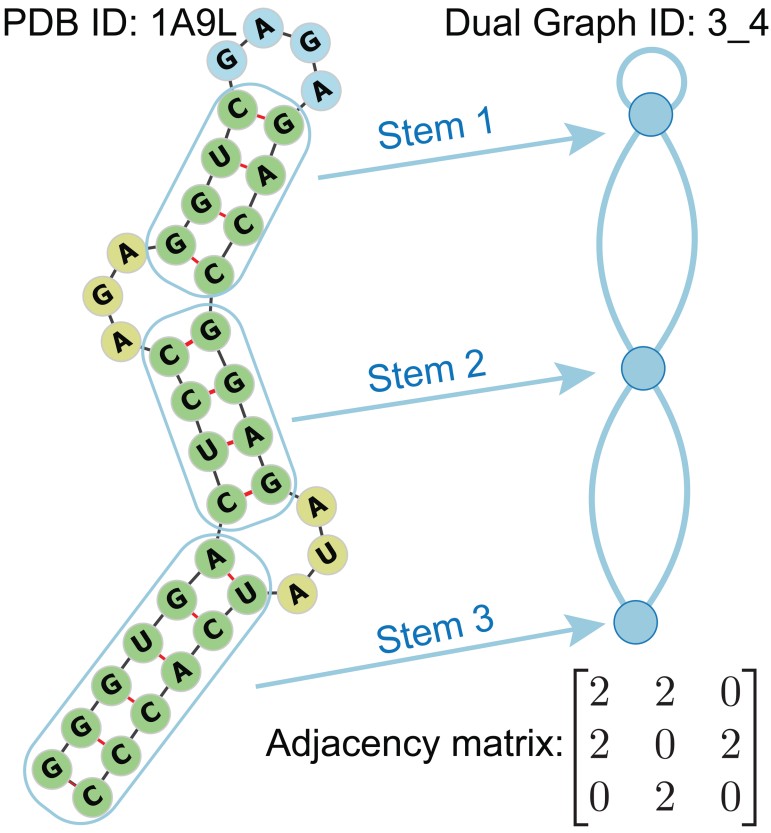

**Fig 1. The secondary structural of 1A9L with its corresponding dual graph representations under "RNA-As-Graphs" (RAG) framework.** The adjacency matrix describes the connectivity between nodes and edges.

Our RAG framework has evolved through multiple versions over time, refining search algorithms to find the most recent existing 3D RNA structures, and applying these tools to important RNA problems, most recently viral RNA frameshifting ([18–21]).

In this work, we use two sets of existing 3D RNA atomic fragments libraries: one reported in 2021 [10] (referred to as the Prior Existing), and the other reported in 2022 later [16] ("Current Existing"). The RNA structures discovered between these two periods can serve as validation data.

## Enumeration of all possible dual graphs

Graph theory enumeration [17] can generate all possible non-isomorphic graph topologies for vertices ranging from 2 to 9, resulting in 110,667 distinct dual graph topologies [22]. Let $G^* = (V^*, E^*)$ be a dual graph with vertices $V^* = \{v_1, v_2, \ldots, v_n\}$ and edges $E^* = \{e_1, e_2, \ldots, e_m\}$. Here $n$ is the number of vertices (or nodes), and $m$ is the number of edges. The adjacency matrix of dual graph $G^*$ is defined as:

$$A = \begin{cases} 2, & \text{if } i = j \text{ and } v_i \text{ has a self-loop} \\ 0, & \text{if } i = j \text{ and } v_i \text{ does not have a self-loop} \\ N_{ij}, & \text{if } i \neq j, \end{cases} \qquad (1)$$

where $N_{ij}$ represents the number of edges between $v_i$ and $v_j$. Since multiple edges between two vertices are allowed. Hence, the minimal value of $N_{ij}$ is 0, and the maximum is 3. In Fig 1, stem 1 has a self-loop; therefore, the entry $A_{11} = 2$. Additionally, there are two edges between stem 1 and stem 2, so the entry $A_{12} = 2$.

To uniquely identify each graph topology, we assign a unique graph ID $a\_k$, where $a$ denotes the number of vertices and $k$ indicates the descending rank based on the Fiedler number of the graph Laplacian within all graph topologies with $a$ vertices. For example, graph ID 4_1 refers to the graph topology with 4 vertices that has the smallest Fiedler number among all 4 vertices graph topologies.

Table 1 lists the numbers of all possible 2D graphs and all graphs corresponding to existing RNA molecules in the Prior Existing and Current Existing. Notably, only a small portion of these graph topologies match known RNA structures, and the remaining motifs are called "hypothetical". This suggests that many graph topologies remain unpaired with real-world RNA structures, highlighting potential areas for future research and discovery. As the

**Table 1. Enumeration of all possible non-isomorphic graph topologies and corresponding RNA molecules in Prior Existing and Current Existing.** The number of all possible graphs, graphs with a match to existing RNA molecules in the Prior Existing and Current Existing corresponding to a specific vertex number is listed.

| Vertex | Possible Graphs | Exist Graphs in 2021 Dataset | Existing in 2022 Dataset |
|---|---|---|---|
| 2 | 3 | 3 | 3 |
| 3 | 8 | 7 | 8 |
| 4 | 29 | 17 | 22 |
| 5 | 110 | 20 | 29 |
| 6 | 508 | 22 | 37 |
| 7 | 2,551 | 21 | 34 |
| 8 | 14,670 | 14 | 20 |
| 9 | 92,788 | 17 | 28 |
| Total | 110,667 | 121 | 181 |

number of vertices increases, the number of possible topologies grows exponentially, with 92,788 topologies for 9 vertices.

To mitigate the effect of unbalanced data, we create datasets based on the number of vertices. Specifically, we gather all possible graph topologies with 4 and 5 vertices into a dataset named "Dataset V4&5." Similarly, we create "Dataset V6" for 6-vertices graphs, "Dataset V7" for 7-vertices graphs, "Dataset V8" for 8-vertices graphs, and "Dataset V9" for 9-vertices graphs. We also compile all topologies into a comprehensive dataset named "Dataset All." We exclude vertices 2 and 3 from further analysis, as they correspond to known RNA structures. These 6 datasets with 4 and more vertices are used in the clustering analysis in Clustering analysis section.

## Topological feature engineering

**Generalization of persistent spectral graph-based features on dual graphs.** We assume that each entry $a_{ij}$ of adjacency matrix $A = (a_{ij})$ of dual graph $G^* = (V^*, E^*)$ represents the distance between vertices $v_i$ and $v_j$ (when $i \neq j$), and $a_{ij}$ is an integer in $[0,1,2,3]$. In this work, we choose the distance-based filtration between two vertices of a dual graph and construct a series of simplicial complexes to form persistent Laplacians. Specifically, given a set of vertices $V^* = \{v_1, v_2, \ldots, v_n\}$ of a dual graph $G^* = (V^*, E^*)$, we consider a nested family of simplicial complexes that are created for a positive real number $d$ (In this work, we set distance filtration parameter $d = 1, 2, 3$). Denoting the simplicial complex generated for $d$ by $K_d$, the traditional $q$th-order Laplacian is the special case of $q$th-order 0-persistent Laplacian at $K_d$:

$$L_q^{d,0} = B_{q+1}^{d,0}(B_{q+1}^{d,0})^T + (B_q^d)^T B_q^d. \tag{2}$$

The spectrum of $L_q^{d,0}$ is simply associated with a snapshot of the filtration,

$$\text{Spec}(L_q^{d,0}) = \{\lambda_{1,q}^{d,0}, \lambda_{2,q}^{d,0}, \cdots, \lambda_{N_q^d,q}^{d,0}\}. \tag{3}$$

With this association, we have the result that the $q$-th 0-persistent Betti number $\beta_q^{d,0} = \beta_q^d$ is equal to the number of zeros in $\text{Spec}(L_q^{d,0})$. For simplicity, we call the $q$-th 0-persistent Betti number as Betti $q$ number. Typically, the Betti $q$ number refers to the number of $q$-dimensional holes on a topological surface. For example, Betti 0, 1, and 2 represent the number of connected components, 1-dimensional cycles (i.e., loops), and 2-dimensional cycles (i.e., cavities) of a given system, respectively. Fig 2 shows the dual graph 4_23 with its corresponding simplicial complexes $K_1$, $K_2$, and $K_3$ when the distance filtration parameter $d = 1, 2$, and 3, respectively. The persistent Laplacians $L_0^{d,0}$ at different distance filtration $d$ of dual graph 4_23 are also shown.

Here, we mainly explore the eigenvalues of 0-th order combinatorial Laplacians (i.e., graph Laplacians) at different frames of a filtration $d$. The reason that we do not utilize the features (i.e., persistent Betti-1 and Betti-2) from higher-order Laplacians is that more than 94% of the Betti-1 and Betti-2 values are zero. Therefore, adding them as features would likely introduce noise rather than a meaningful signal, potentially compromising model performance. Specifically, we chose 18 statistical values of the $\text{Spec}(L_q^{d,0})$ features to describe a dual graph as follows:

- 4 features from $\text{Spec}(L_0^{d,0})$ when $d = 1$: summation, non-zero smallest, variance, and the number of zeros;

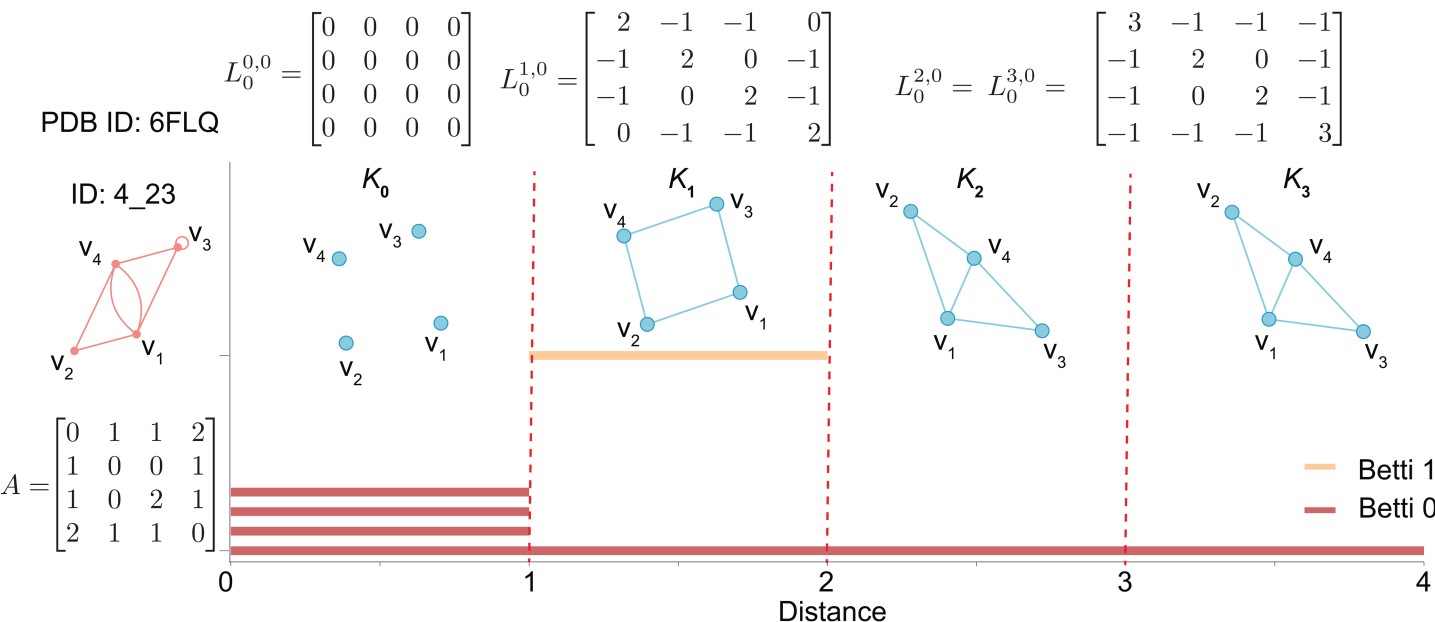

**Fig 2. Adjacency matrix of dual graph 4_23 (PDB ID 6FLQ) and its simplicial complexes and persistent Laplacians at different distance filtration values.** Here, the yellow line represents the barcode of Betti 1, while the red lines are the barcode of Betti 0. (See S1.1 in S1 Methods for details on Betti numbers.)

- 7 features from $\mathrm{Spec}(L_0^{d,0})$ when $d = 2$: summation, non-zero smallest, maximum, average, standard deviation, variance, and the number of zeros;
- 7 features from $\mathrm{Spec}(L_0^{d,0})$ when $d = 3$: summation, non-zero smallest, maximum, average, standard deviation, variance, and the number of zeros.

However, the distance filtration does not consider the self-loop information in the adjacency matrix $A = (a_{ij})$ when $i = j$. Therefore, we include the average degree (Avg(Deg)) as an additional feature (the 19th feature) to describe a dual graph, which is defined as:

$$\mathrm{Avg(Deg)} = \frac{\sum_i a_{ii}}{2n}, \qquad (4)$$

where $n$ is the number of vertices (i.e., $|V^*|$). Basic concepts regarding persistent spectral graphs are described in S1.1 and S1.2 of S1 Methods.

**Dual-RAG-IF design setup.** We use the Dual-RAG-IF package [18,19,23] to find RNA sequences that fold onto a target 2D fold described as a dual graph. The initial RAG-IF program is based on a genetic algorithm to mutate an initial sequence, determine its 2D folding at each step, and then generate collectively a large number of sequences that fold on the desired motif, which are then sorted by minimal mutations from the starting sequence. More details are available in [23]. Dual-RAG-IF has successfully applied to find structure-altering mutations for the SARS-CoV-2 frameshifting element [18,19].

There are three key steps in Dual-RAG-IF: **1) Mutation region identification**: Mutation regions can be determined either manually or automatically. For manual design, the input must include a target 2D structure in dot-bracket notation defining with the target dual graph, along with a sequence that marks mutation regions by representing residues as 'N'. For automatic design, we only need the target dual graph information, and the entire sequence is

taken as the mutation region. 2) **Candidate sequence generation**: Candidate sequences with mutations are generated using a genetic algorithm. Each candidate is evaluated by applying two 2D folding programs (such as NUPACK[24], IPknot[25], and PKNOTS[26]) capable of handling pseudoknots to verify the resulting fold, and a sequence is considered successful when both programs agree on the target fold. In our application, we use the NUPACK and IPknot packages. 3) **Sequence optimization**: The candidate sequences are ordered by minimal mutations, ensuring an optimal final design.

## Results

### Clustering analysis

**Overview.** Our two datasets from 2021 and 2022 contain graph topologies. Graphs corresponding to *existing* 3D RNA structures are assigned the label of 1. The remaining hypothetical RNA topologies can be either non-RNA structures (label 0) or real RNAs (label 1), but their true nature is currently unknown due to the limited number of discovered RNA molecules. Our aim is to identify the highly RNA-like structures among these hypothetical RNA topologies. To address this, unsupervised machine learning methods are most suitable.

Specifically, we apply 6 clustering algorithms that can partition all possible graphs into exactly two clusters: $k$-means clustering, mini-batch $k$-means clustering, Gaussian mixture models, hierarchical (ward) clustering, spectral clustering, and birch clustering. PSG-based features were used as inputs for these clustering methods. The cluster containing the majority of existing real RNAs was designated as the RNA-like cluster (graphs in this cluster were assigned a predicted label of 1). The other cluster was designated as the non-RNA-like cluster (graphs in this cluster were assigned a predicted label of 0). The closer an RNA motif is to the RNA-like cluster center, the higher the possibility of being the most RNA-like graph. This approach leverages the existing RNA structures to infer the potential nature of the hypothetical graph topologies.

Our 19 PSG-based features describe each graph topology and are used in the six clustering algorithms across datasets grouped by vertex numbers: Dataset V4&5, Dataset V6, Dataset V7, Dataset V8, Dataset V9, and Dataset All for comparison. With unsupervised machine learning clustering algorithms, we have no predefined labels for hypothetical graphs to compare the predictions against, but we know the label of existing graph topologies (label 1). Therefore, we primarily use sensitivity to evaluate the clustering performance. Sensitivity refers to the percentage of existing graph topologies correctly clustered into the RNA-like cluster. Therefore, a high sensitivity score indicates that the clustering method is accurately capturing and grouping the RNA-like graph topologies, reflecting good prediction performance. Additionally, we also use the silhouette score, which measures how similar each graph is to its assigned cluster (cohesion) compared to the other clusters (seperation), and the homogeneity score, which measures how uniformly the clustering results group the existing RNA structures within the same cluster. High silhouette scores and high homogeneity values indicate effective clustering.

**Comparative analysis of clustering methods across datasets.** To better visualize the clustering results, we use principal component analysis (PCA) to reduce the feature dimension to 2. Fig 3 presents the clustering results of six different algorithms (columns in Fig 3) applied to various datasets (rows in Fig 3), characterized by persistent spectral graph-based (PSG) features. In Fig 3, all graphs in the Prior Existing are visualized with dots, while all cross symbols represent the hypothetical graph topologies. The red cluster represents the RNA-like cluster, while the grey cluster indicates the non-RNA-like cluster. Due to the large

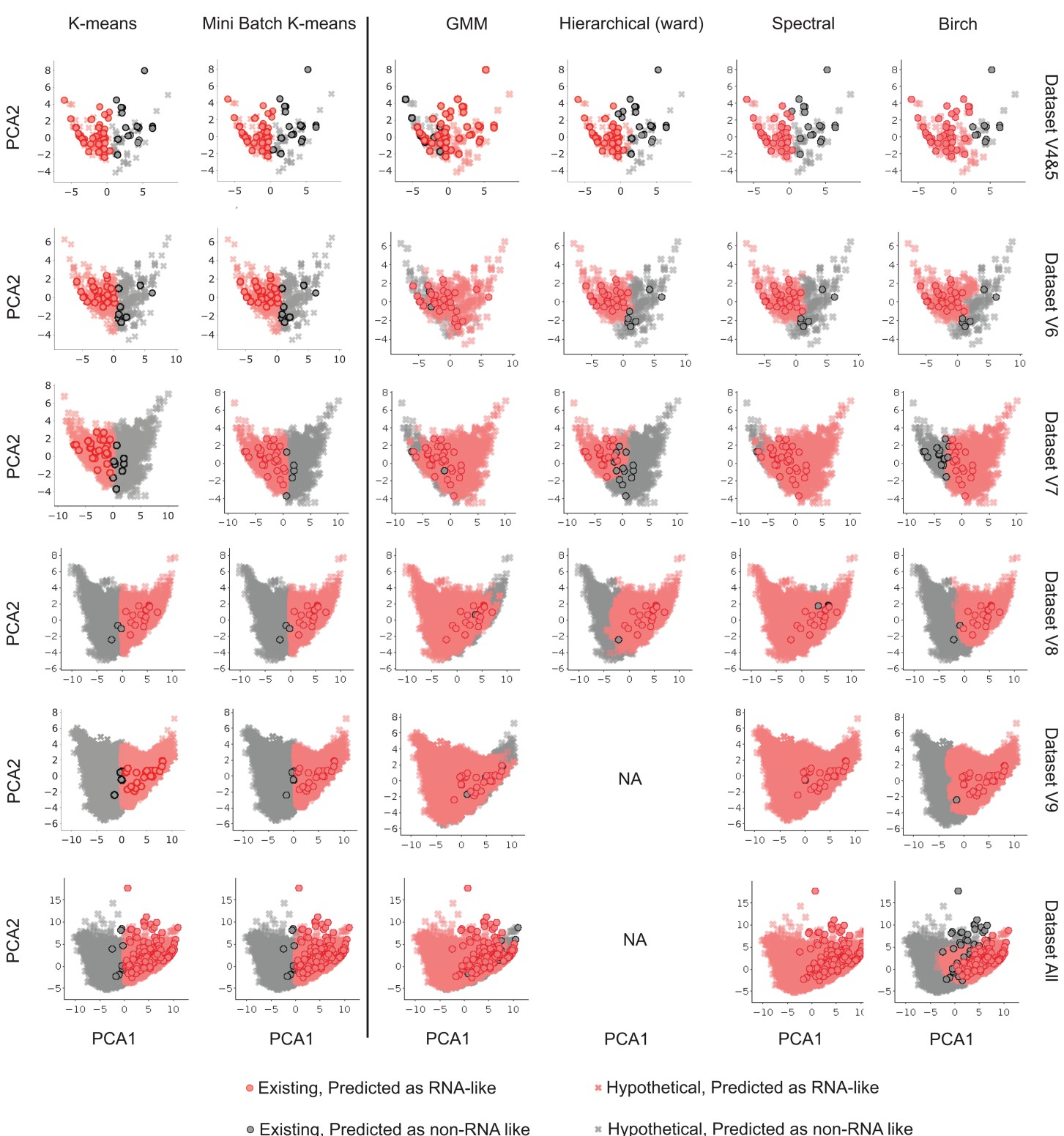

**Fig 3. Clustering results by six methods with persistent spectral graph-based features on six datasets grouped by motif vertices: Dataset V4&5, Dataset V6, Dataset V7, Dataset V8, Dataset V9, and Dataset All.** The red and grey clusters are the RNA-like and non-RNA-like clusters, respectively. The circled red dot symbols indicate existing RNA molecules, while the cross symbols denote hypothetical graphs. The first two methods (k-means) exhibit the most reliable performance.

time complexity of hierarchical (ward) clustering on larger datasets V9 and All, we did not report clustering performance for these datasets (NA in Fig 3). Thus the first two $k$-means methods (the $k$-means clustering is also used in our previous RNA clustering work [10]) exhibit good performance and scale well as dataset size increase.

We see that the $k$-means clustering and mini-batch $k$-means clustering have well-separated clusters across all datasets. The GMM method generally has fewer mis-clustered grey dots, but its two clusters tend to overlap more compared to other clustering methods. Hierarchical clustering with Ward's method performs well especially as datasets increase but is not feasible for larger datasets due to its $O(n^3)$ time complexity [27]. Spectral clustering shows varied performance on different datasets; it clusters every existing graph into the RNA-like cluster on Dataset V9 and Dataset All but produces an almost non-existent non-RNA-like cluster. Lastly, Birch clustering shows good separation for smaller datasets, but its performance appears to degrade on Dataset All. The sensitivities and percentages of graphs clustered as RNA-like are given in Table 2. In particular, we see from Table 2 that the predicted percentage of RNA-like motifs among all clustering methods on Dataset All varies from 32.505% to 99.990%. The more reliable $k$-means and mini-batch $k$-means clustering methods based on combined sensitivity, silhouette, and homogeneity scores yield about 46%. Because our analysis consider lower vertex numbers, namely up to 9, this estimate is likely a lower bound.

We also notice that the clustering performance tends to be poorer on smaller datasets compared to larger ones. This is because smaller datasets share more similar topological and geometric features, and these highly skewed feature distributions have negative impact on the performance of machine learning algorithms.

We also analyze the performance of each clustering method by its silhouette score and homogeneity score (See Fig A in S1 Methods). The silhouette score measures how similar a data point is to its own cluster compared to other clusters, indicating the cohesion and separation of clusters. The homogeneity score assesses whether clusters contain only data points that belong to a single ground truth class, reflecting the purity of clusters with respect to true labels. Higher silhouette and homogeneity scores indicate a more effective clustering algorithm with well-separated and accurately defined clusters. Although the spectral clustering algorithm achieves 100% sensitivity on Dataset V9 and Dataset All, it has a relatively low silhouette score and homogeneity score, indicating poor clustering performance on these two

**Table 2. Comparison of sensitivity and percentage of RNA-like graphs under six clustering methods. The red values denote predictions by the $k$-means and mini-batch $k$-means methods of the RNA-like group size.**

| | | Datasets | | | | | |
|---|---|---|---|---|---|---|---|
| | | V4&5 | V6 | V7 | V8 | V9 | All |
| $k$-means | Sensitivity(%) | **72.973** | **86.364** | 85.714 | **92.857** | **94.118** | **97.297** |
| | RNA-like(%) | 59.712 | 55.118 | 49.079 | 48.712 | 46.241 | 46.017 |
| Mini-batch $k$-means | Sensitivity(%) | 56.757 | **86.364** | **95.238** | **92.857** | **94.118** | **97.297** |
| | RNA-like(%) | 42.446 | 56.496 | 59.153 | 48.541 | 42.683 | 46.795 |
| GMM | Sensitivity(%) | 86.486 | 90.909 | 90.476 | 78.571 | 94.118 | 90.090 |
| | RNA-like(%) | 76.259 | 72.638 | 72.677 | 72.904 | 73.517 | 73.412 |
| Hierarchical (ward) | Sensitivity(%) | 67.568 | 86.364 | 71.429 | 100.000 | NA | NA |
| | RNA-like(%) | 61.151 | 60.236 | 33.242 | 69.659 | NA | NA |
| Spectral | Sensitivity(%) | 70.270 | 86.364 | 90.476 | 78.571 | 100.000 | 100.000 |
| | RNA-like(%) | 58.993 | 60.630 | 90.396 | 99.843 | 98.189 | 99.990 |
| Birch | Sensitivity(%) | 83.784 | 86.364 | 57.143 | 85.714 | 94.118 | 76.577 |
| | RNA-like(%) | 85.612 | 68.504 | 29.008 | 48.480 | 62.461 | 32.505 |

datasets. Similarly, while GMM has the highest sensitivities on Datasets V4&5 and V6, its silhouette and homogeneity scores are the lowest among the clustering methods evaluated. This suggests that relying solely on sensitivity to choose the suitable clustering methods for identifying top RNA-like graphs may not be sufficient. Therefore, a more comprehensive approach, considering multiple evaluation metrics and clustering algorithms are necessary to assess their performance and hence prediction regarding most likely RNA-like graphs.

To assess cluster separability and methodological robustness, we also evaluate our clustering results using multiple complementary approaches. First, we employ UMAP embeddings [28] to validate whether the RNA-like/non-RNA-like separation persists beyond linear assumptions. The 2D UMAP visualization (see Fig B in S1 Methods) demonstrates clear geometric separation between two clusters with minimal overlap and absence of unimodal distribution patterns that would invalidate binary clustering. Second, we further verify that our data have a multimodal distribution by applying two statistical tests, Hartigan's dip test[29] and Silverman's test [30] (See Fig C in S1 Methods). These two tests emphasize that all features used in the clustering have a small dip value with p-value less than 0.05, and the number of peaks is greater than 1, indicating that our data are multimodal and therefore the 50% RNA-like result stems not from the $k$-means applied to a unimodal distribution.

**Comparison of different feature performance of $k$-means clustering.** For the $k$-means clustering method, we also evaluate in Table 3 the $s$ and $e$ features as used before [10] compared to the topological and geometric features by PSG introduced in this work. We see that the sensitivity (the percentage of existing graph topologies in 2021 dataset correctly clustered into the RNA-like cluster) has been improved by PSG features from 88.288% to 97.297% on Dataset All.

## Top central structures within clusters among six clustering methods

Now it is interesting to explore the most RNA-like and unlikely RNA-like predicted hypothetical graphs. For each dataset (V4&5, V6, V7, V8, V9, and All), we selected the top 15 RNA-like hypothetical graphs closest to the center of the RNA-like cluster and the top 15 non-RNA-like hypothetical graphs closest to the center of the non-RNA-like cluster using each of the six clustering methods. We then collect graphs that commonly appeared in the top 15 lists of three, four, and five clustering algorithms, for both RNA-like and non-RNA-like structures. These motifs are considered to be the most likely RNA-like 2D structures.

Fig 4 illustrates the most RNA-like (red) and non-RNA-like (grey) graphs that commonly appear five times among all clustering algorithms among all datasets (V4&5 - V9), respectively. They are all 4, 5, and 6-vertex graphs. While the top RNA-like motifs appear to be combinations of two different subgraphs, the top non-RNA-like motifs resemble clusters of polygons.

**Table 3. Comparison of sensitivity and percentage of RNA-like graphs using different features for $k$-means clustering.**

|  |  | Datasets | | | | | |
|---|---|---|---|---|---|---|---|
|  |  | V4&5 | V6 | V7 | V8 | V9 | All |
| $k$-means $s$ and $e$ | Sensitivity(%) | 70.270 | 72.727 | 57.143 | 64.286 | 64.706 | 88.288 |
|  | RNA-like(%) | 70.504 | 67.913 | 71.815 | 69.843 | 71.749 | 72.169 |
| $k$-means PSG | Sensitivity(%) | **72.973** | **86.364** | **85.714** | **92.857** | **94.118** | **97.297** |
|  | RNA-like(%) | 59.712 | 55.118 | 49.079 | 48.712 | 46.241 | 46.017 |

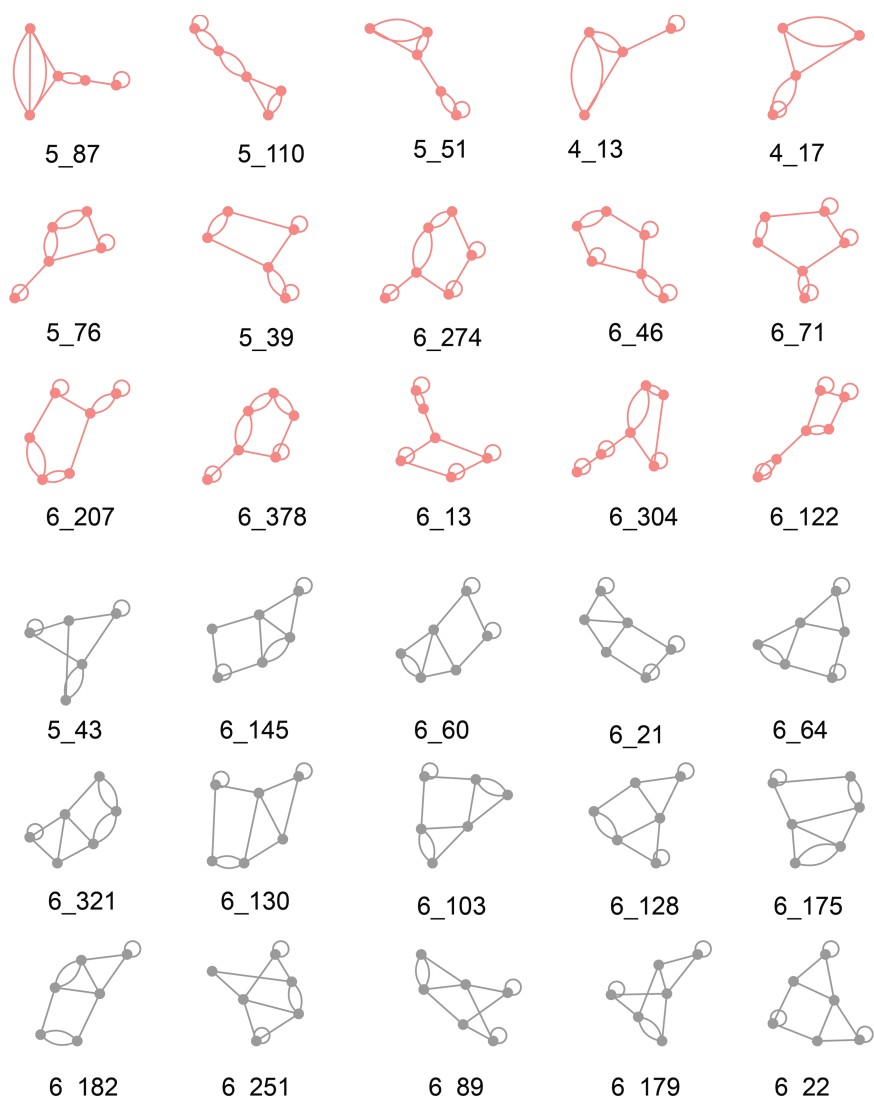

**Fig 4. Graphs commonly clustered as top RNA-like and non-RNA-like graphs by five clustering methods.** Red graphs represent the RNA-like graphs, while grey graphs represent the non-RNA-like graphs.

We also partition the most likely RNA-like and non-RNA-like graphs (from Fig 4) into subgraphs by the algorithms described in [31]. Tables 4 and 5 list the subgraphs of the top 15 most likely RNA-like graphs and the top 15 non-RNA-like graphs, respectively. Surprisingly, we find that the top RNA-like graph topologies all have corresponding subgraphs, while the top 15 non-RNA-like graphs are non-separable. Recall that subgraph division preserves junctions and pseudoknots [22]. This suggests that RNAs favor hierarchical combinations of simple motifs rather than complex intertwined topologies. This striking result may guide future RNA design. Not only can we suggest which topologies are RNA-like, but also how to design them as simple combinations of subgraphs. In particular, the top 15 candidates for novel RNA motifs include graphs (4_17, 5_76, 5_39, and 6_71) that were indeed found in the 2022 RNA dataset.

**Table 4. Top 15 most RNA-like dual graph IDs and their subgraphs.** Here, we use blue IDs to represent the existing RNA motifs. The three graph IDs colored red are those we design in the last subsection of Results.

| Graph ID | Subgraphs | Graph ID | Subgraphs | Graph ID | Subgraphs |
|---|---|---|---|---|---|
| 5_87 | 2_1, 2_2, 3_2, 3_7, 4_18 | 6_46 | 2_2, 5_18 | 5_110 | 2_2, 3_4, 3_6, 4_17 |
| 6_71 | 2_2, 5_18 | 5_51 | 2_1, 2_2, 3_2, 3_8, 4_13 | 6_207 | 2_2, 5_55 |
| 4_13 | 2_1, 3_8 | 6_378 | 2_1, 5_83 | 4_17 | 2_2, 3_6 |
| 6_13 | 2_1, 2_2, 3_2, 4_19, 5_4 | 5_76 | 2_1, 4_26 | 6_304 | 2_1, 3_1, 4_26, 5_76 |
| 5_39 | 2_2, 4_20 | 6_274 | 2_1, 5_55 | 6_122 | 2_1, 2_2, 3_2, 4_20, 5_37 |

**Table 5. Top 15 most non-RNA-like dual graph IDs their subgraphs.**

| Graph ID | Subgraphs | Graph ID | Subgraphs | Graph ID | Subgraphs |
|---|---|---|---|---|---|
| 5_43 | / | 6_145 | / | 6_60 | / |
| 6_21 | / | 6_64 | / | 6_321 | / |
| 6_130 | / | 6_103 | / | 6_128 | / |
| 6_175 | / | 6_182 | / | 6_251 | / |
| 6_89 | / | 6_179 | / | 6_22 | / |

## Topological analysis on top RNA-like and non-RNA-like structures

For each graph, we can calculate its persistent betti 0 ($\beta_0^{0,d}$) and persistent betti 1 ($\beta_1^{0,d}$) based on distance filtration. In Fig 5, we analyze the topological barcode for top RNA-like graphs 5_51 and 4_13, as well as top non-RNA-like graphs 6_145 and 6_251. Topological barcode measures the persistence of topological features like connected components, loops, and voids within a dataset across varying scales. The persistent betti 0 ($\beta_0^{0,d}$) and persistent betti 1 ($\beta_1^{0,d}$) can be obtained by counting the red and yellow bars at a specific distance filtration, respectively. Two patterns from Fig 5 can be observed: 1) RNA-like graphs exhibit more changes in Betti 0s, whereas non-RNA-like graphs do not show many Betti 0 changes. 2) RNA-like graphs do not have Betti 1 bars, while non-RNA-like graphs are more likely to have at least one Betti 1 bar. The same patterns exist in all graphs illustrated in Fig 4. Details can be found in the github repo PSGRNAClustering. Therefore, we suggest that future research can follow these rules in designing new RNA-like motifs.

Fig 6 shows average Betti number curves along various filtration parameters. From Fig 6a) and Fig 6d), we can see that the average Betti-1 value remains below 1 for all existing RNA

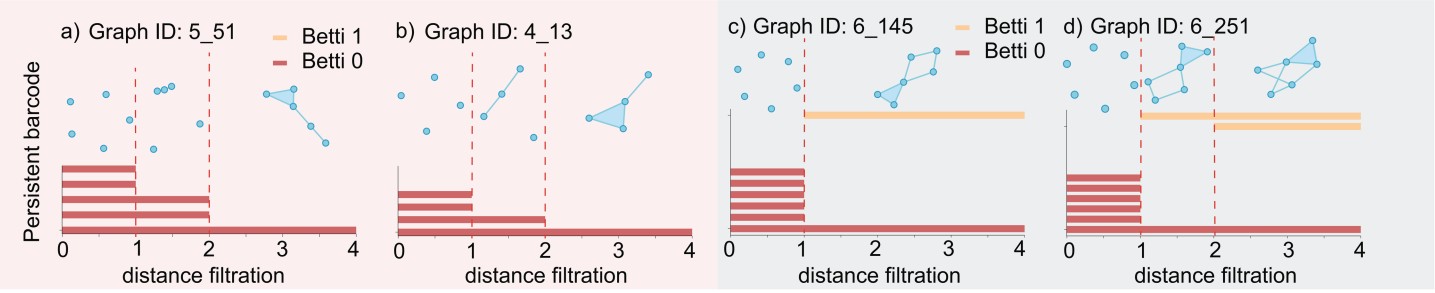

**Fig 5. Persistent barcode of graph topologies.** The red panel shows the persistent barcode of RNA-like graphs a) 5_51 and b) 4_13. The grey panel shows the persistent barcode of non-RNA-like graphs c) 6_145 and d) 6_251. At a specific distance filtration $d$, the count of red bars represents the persistent betti 0 $\beta_0^{0,d}$, while the count of yellow bars indicates the persistent betti 1 $\beta_1^{0,d}$.

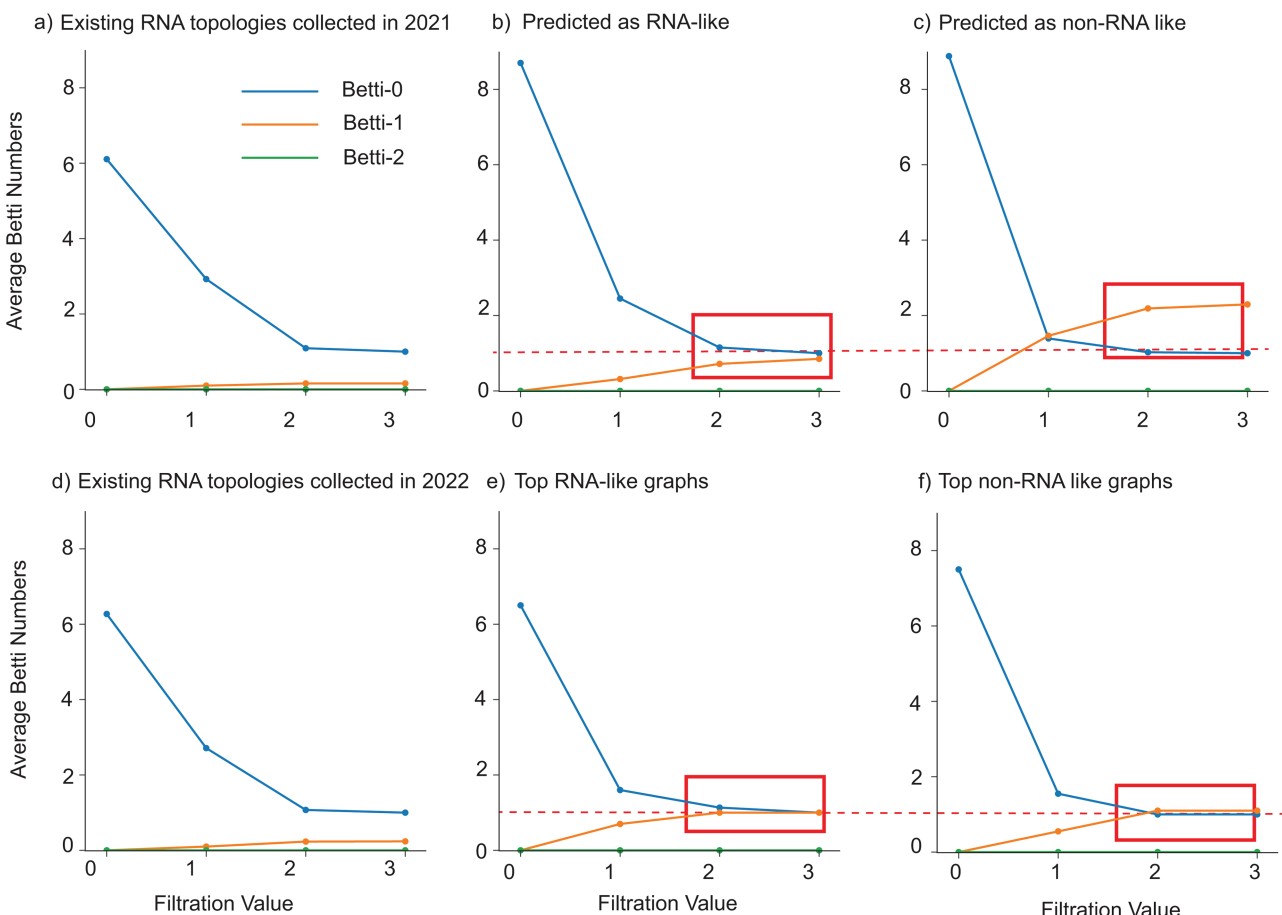

**Fig 6. Average Betti number curves along different filtration parameters using data from a) previously collected existing RNA topologies in 2021, b) predicted RNA-like graphs, c) predicted non-RNA-like graphs, d) previously collected RNA topologies in 2022, e) top RNA-like graphs, and f) top non-RNA-like graphs.** The blue, orange, and green curves correspond to Betti-0, Betti-1, and Betti-2, respectively.

topologies collected in 2021 and 2022. This observation suggests that existing RNAs are not prone to forming loops under all filtration conditions. Moreover, although the average Betti-1 value for the predicted RNA-like graphs in Fig 6b) exceeds that of the existing RNA topologies, we can still notice a substantial difference between the predicted RNA-like and predicted non-RNA-like graphs. Predicted non-RNA-like graphs exhibit a higher tendency to form loops compared to RNA-like graphs, which aligns with the findings illustrated in Fig 5. Furthermore, similar Betti-1 patterns exist for top RNA-like and top non-RNA-like graphs: top RNA-like graphs have a relatively lower average Betti-1 number compared to top non-RNA-like graphs.

## Validation

Using the 121 existing RNA dual graphs collected in 2021, we apply the $k$-means clustering method with PSG features to partition all possible dual graphs into two clusters: an RNA-like cluster and a non-RNA-like cluster. In the RNA-like cluster, 10 4 or 5-vertex, 17 6-vertex, 15 7-vertex, 14 8-vertex, and 12 9-vertex dual graphs are found in the new 2022 dataset. These

results indicate the reliability of our model in accurately separating dual graphs into RNA-like and non-RNA-like.

## RNA motif design examples

Using the subgraph information in Table 4 and leveraging our previous work with the Dual-RAG-IF package, we have successfully designed several high-likelihood RNA-like motifs from their subgraphs for graphs: 4_13, 4_17, and 5_39 as shown in Fig 7. Specifically, graph 4_13 contains two subgraphs that align with existing RNA motifs: 2_1 and 3_8. For subgraph 2_1, we select the RNA motif with PDB ID 1K2G, while for subgraph 3_8, we chose the RNA motif with PDB ID 1F5U. We then combine the RNA sequences of these two motifs and use them as input for the inverse folding Dual-RAG-IF package [18,19,23]. Through nucleotide mutations by a genetic evolution algorithm, we generate 555 sequences whose graph IDs match 4_13. Similarly, highly likely RNA topology 4_17 has two subgraphs that align with existing RNA motifs: 2_2 and 3_6. For subgraph 2_2, we select the RNA motif with PDB ID 1A3M, while for subgraph 3_6, we chose the RNA motif with PDB ID 1YZ9. By combining these sequences together and feeding into the Dual-RAG-IF, we obtain 117 potential RNA sequences that are predicted to fold onto our target graph ID 4_17. In addition, 84 RNA sequences corresponding to graph ID 5_39 are also designed similarly from PDB structures 6AZ3 and 4RZD for subgraphs 2_2 and 4_20, respectively. Fig 7 lists 60 RNA sequences corresponding to these

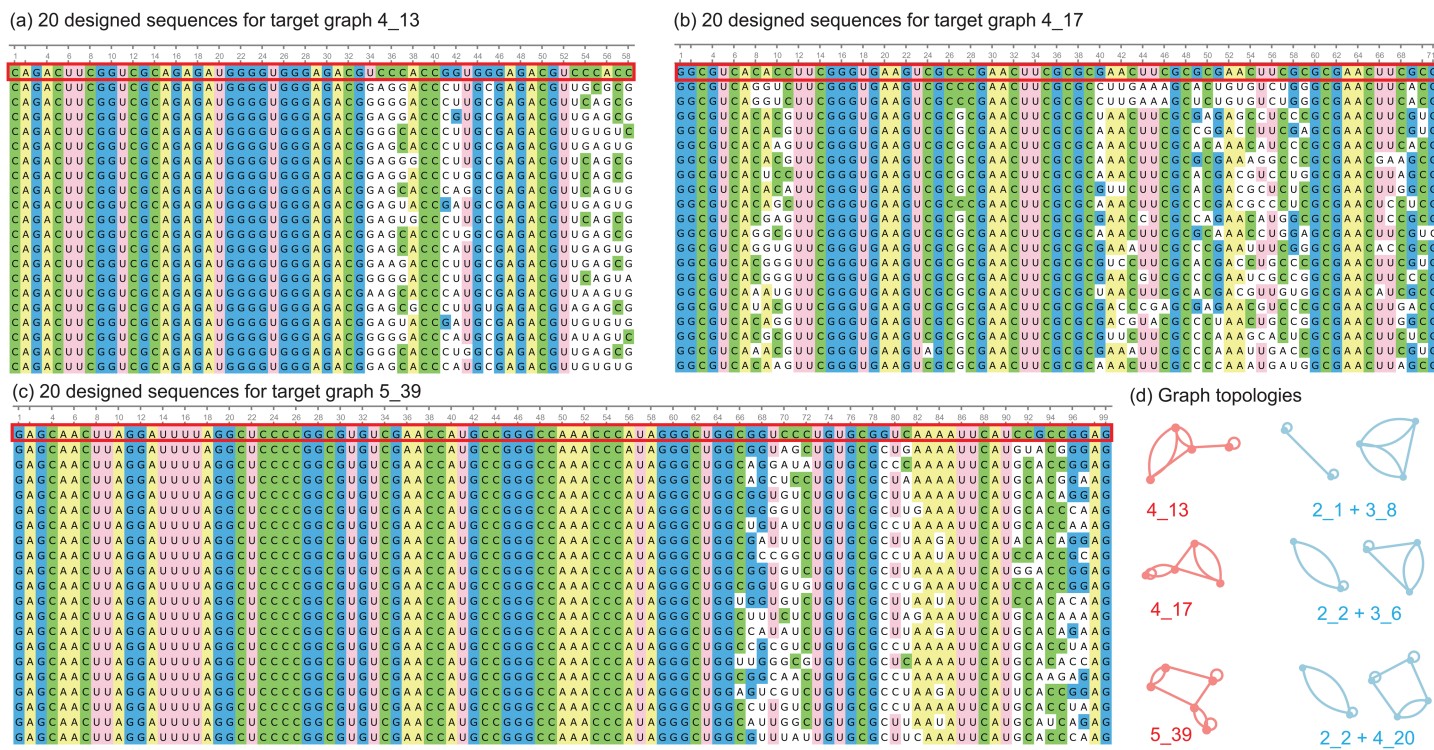

**Fig 7. Multiple sequence alignment plot generated by Unipro UGENE [32].** (a), (b), and (c) shows 20 designed RNA sequences for target graphs 4_13, 4_17, and 5_39, correspondingly. The first sequence (in red box) in each MSA panel is a combined sequence from two subgraphs. (d) shows the target graph topologies (red) and their subgraphs (blue).

three target motifs 4_13, 4_17, and 5_39. Detailed sequence information can be found in the designs folder of the github repo PSGRNAClustering.

## Summary and discussion

### Summary

In this work, we have introduced new topological descriptors (PSG) and showed their value in clustering the library of RNA motifs trained by existing RNAs to predict RNA-like hypothetical topologies. We show that the $k$-means clustering method performs especially well among all six clustering methods considered, as suggested by its high sensitivity, silhouette, and homogeneity scores. The predicted RNA-like topologies of around 46% suggest that many possible RNA motifs in the RNA motif atlas are likely designable or exist in nature; this estimate of the size of the RNA motif universe will be refined as our solved RNA database increases in volume. In particular, our striking results indicate that all top RNA-like motifs are decomposable into subgraphs, whereas our top non-RNA-like topologies are irreducible, underscores the modularity and hierarchical nature of biological RNAs. Our work also directly suggests how to design these new RNA-like topologies by combining sequences that correspond to their subgraphs in a build-up strategy as done previously [7,23].

### Limitations

While our study presents a powerful framework for RNA motif design, clearly less frequently observed conformations may be misclassified as non-RNA-Like due to overfitting to a limited existing dataset. That is why we frequently update our classifications and correct any database errors that have occurred [9,10,17,22,31,33–35].

The RAG approach has evolved significantly since the initial development of RNA-as-Graph (RAG) in 2003, where we first introduced tree-graph representations for RNA motifs and dual graphs for pseudoknots [17]. As structural data have increased, we enhanced the RAG resource in 2011 by incorporating additional RNA structures and developing new analytical tools for substructure identification and graph searching [33]. In 2014, we developed a tree graph partitioning using Fiedler vectors for the discovery of RNA modularity [31], and introduced RAG 3D tree graphs to characterize global helical arrangements in large RNA structures [34]. Building on these developments, we expanded the RAG framework in 2015 to handle larger tree graphs (up to 13 vertices, 260 nucleotides) while implementing Partitioning Around Medoids clustering to better classify "RNA-like" motifs from hypothetical structures [35]. We also presented RAG-3D, a dataset of RNA tertiary (3D) structures and substructures with a web-based search tool, to enable comprehensive graph-based searches of RNA tertiary structures and substructures [9].

In 2019, enhanced dual graph enumeration algorithms were introduced for generating libraries of dual graph topologies for 2-9 vertices, along with an improved algorithm for RNA subgraph identification [22]. Most recently, in 2021, we incorporated Fiedler vector-based feature design and scoring models to further refine our graph candidate selection process [10]. Many successful design and conformational analysis applications to the SARS-CoV-2 frameshifting viral element were made possible due to the compact graph representation [18,19,21,36,37].

Because coarse-grained graph representations are not suitable for capturing all-atom conformational and dynamic fluctuations important for RNA in vivo, all-atom modeling is essential for a complete picture (e.g., [21,36]). Ultimately, experimental validations are needed [18,37].

### Future directions for RNA motif design

Our work indicates that the RNA-like universe is at least 46% and that biological RNA topologies are more likely to contain subgraphs that distinguish them from non-RNA-like structures. Echoing this finding from the perspective of Persistent Spectral Graph (PSG) analysis is that RNA-like graphs tend to exhibit more changes in Betti 0 numbers or the number of connected components, and show an absence of Betti 1 bars or 1-dimensional loops. These findings suggest that the topological features captured by PSG, such as Betti numbers, can be critical indicators of RNA-like characteristics.

Based on these insights, it would be promising to build a comprehensive library of RNA-like graphs characterized by their topological and spectral properties, such as their Betti number profiles and subgraph compositions. This library could serve as a valuable resource for RNA motif design by providing a structured database of potential RNA structures that are more likely to exhibit stable and functional conformations. Researchers could use this library to efficiently screen for novel RNA motifs with desired structural and functional properties, potentially guiding the discovery of new RNA-based therapeutics, biosensors, and regulatory elements. Given the rising popularity and success of AI prediction for systems (such as proteins) where databases are extensive, maintenance of such databases for RNA remains crucial.

In conclusion, our methods and findings provide valuable tools and topological insights for guiding future RNA motif design by highlighting the importance of subgraph patterns and topological features in distinguishing RNA-like and non-RNA-like structures. Our work points to successful build-up approaches for combining nucleotide sequences of corresponding subgraphs of the target graph, as done using RAG [23]. Our library of RNA-like graphs, informed by PSG analysis and enhanced by machine learning, helps the discovery and development of novel RNA motifs, paving the way for innovative RNA-based technologies.

### Supporting information

**S1 Methods. Supplementary methods.** This section includes the definition of basic topological concepts (S1.1), persistent spectral graphs (S1.2), clustering algorithms (S1.3), and evaluation metrics (S1.4).
(PDF)

### Author contributions

**Conceptualization:** Rui Wang, Tamar Schlick.

**Data curation:** Rui Wang.

**Formal analysis:** Rui Wang.

**Funding acquisition:** Rui Wang, Tamar Schlick.

**Investigation:** Rui Wang, Tamar Schlick.

**Methodology:** Rui Wang, Tamar Schlick.

**Resources:** Tamar Schlick.

**Software:** Rui Wang.

**Supervision:** Tamar Schlick.

**Validation:** Rui Wang, Tamar Schlick.

**Visualization:** Rui Wang.

**Writing – original draft:** Rui Wang.

**Writing – review & editing:** Tamar Schlick.

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
