## [Decision Letter · Decision Letter 0]

18 Mar 2025

PCOMPBIOL-D-25-00058

How Large is the Universe of RNA-Like Motifs? A Clustering Analysis of RNA Graph Motifs Using Topological Descriptors

PLOS Computational Biology

Dear Dr. Wang,

Thank you for submitting your manuscript to PLOS Computational Biology. After careful consideration, we feel that it has merit but does not fully meet PLOS Computational Biology's publication criteria as it currently stands. Therefore, we invite you to submit a revised version of the manuscript that addresses the points raised during the review process.

Please submit your revised manuscript within 60 days May 18 2025 11:59PM. If you will need more time than this to complete your revisions, please reply to this message or contact the journal office at ploscompbiol@plos.org. Please include the following items when submitting your revised manuscript:

We look forward to receiving your revised manuscript.

Kind regards,

Mingfu Shao, Ph.D.

Academic Editor

PLOS Computational Biology

Ilya Ioshikhes

Section Editor

PLOS Computational Biology

**Journal Requirements:**

At this stage, the following Authors/Authors require contributions: Rui Wang. Please ensure that the full contributions of each author are acknowledged in the "Add/Edit/Remove Authors" section of our submission form.

5) Thank you for stating "The code and data for the feature and clustering algorithms are available at the public repository PSGRNA-Clustering. The RNA inverse folding using dual graph representations package is available at Dual-RAG-IF."  Please provide direct links in the online submission form to access the datasets. 

**Reviewers' comments:**

Reviewer's Responses to Questions

Reviewer #1: This work explores the space of all possible dual graphs (a coarse-grained representation of RNA structures) with number of vertices ranging from 4 to 9. These structures are clustered into two groups which are later labeled as RNA-like or not-RNA-like using a handful of known RNA structures. Several commonly used clustering algorithms were tried using statistical features generated from the eigenvalues of the graph Laplacians on a sequence of subgraphs (thresholded by edge weights). The study suggests that ~46% of the graphs could correspond to RNA structures. It also provides a landscape for designing novel RNA motifs. I mainly have questions on the (1) reliability of the main conclusion and (2) utilizing the problem setup (small number of features used with interpretable methods) to include interpretations of the model.

1.According to the description in 2.2.1, it seems that “persistence” is never used. A simpler descriptio without unnecessary jargon would make it more readible, especially for readers who are not familiar with the concept of persistence. For example, “exploring the eigenvalues of combinatorial Laplacians at different frames of a filtration.”

2.Related to the previous point, there is also no need to include the parameter $q$ and $q$th-order persistent Laplacian, as only standard graph Laplacians are used.

3.The interpretation of the main result could use more clarification. I have the following questions: (1) It is hard to tell from the scatter plots in Figure 3 whether there is separation between the two clusters. The method of choice, K-means, works well on datasets with clear separation between clusters. For example, if the dataset is drawn from a 1D Gaussian, k-means with k=2 will just divide the dataset into two, one below the mean and one above the mean. The ~50% RNA-like and the significantly different percentage from GMM makes me wonder if the ~50% is due to applying K-means to a unimodal distribution. It would be helpful to also visualize the distribution as density plots and potentially in different embeddings, such as UMAP, etc. (2) Conceptually, it is possible for the ground-truth to be, for example, 20% of RNA-like structures. However, the current validation can not distinguish between the two (vs. the predicted 46%). It would be helpful to include the additional metrics (like homogeneity scores) in the performance table in the main manuscript for more comprehensive assessment of the performance. Also, is there any resource, direct or indirect, on known negative samples? If such results exists, including specificity could be very helpful to clarify this point. In addition, if some negative samples are available, it is also interesting to explore semi-supervised learning which may outperform of the current approach where separation of data and label assignment are done in two sequential steps.

4.I find the interpretation in 3.3 interesting. Why is Betti-1 used here but not in the clustering analysis (2.2.1)? It would be interesting to see whether the patterns identified from a few example graphs (Figure 4) apply to the entire dataset. Can you visualize the average Betti curves across all graphs of each cluster? Some statistical tests for distinguishing the curves from the two groups are useful to formally state the observation. Additionally, interpretations of the 18 features should be included and presented in a similar manner.

5.Can the authors verify if the subpanels of Figure 3 are correct? At least for the top left panel, it looks like much more than 139 points (total possible graphs for V4&5).

6.2.2.1 states 18 features while 3.1.1 states 19 features.

7.I appreciate the property summary of dual graphs in 2.1.1. Can you elaborate more on how the listed properties (non-surjective, non-injective) affect the implication of the clustering analysis results?

Lastly, I am curious about why the performance is generally worse on smaller graphs? Could including higher-order Laplacians lead to improvements?

Reviewer #2: This study employs clustering analysis using graph data mining techniques to represent RNA structures through an "RNA-Like" graph-topological framework. However, several concerns arise:

1. Graph Representation of RNA Structures

A. The authors treat RNA structures as graphs, with double-stranded helical stem regions represented as nodes. However, considering all base-pairing regions as equivalent vertices may introduce bias in the graphical representation of RNA structures. For instance, a short 5-nt stem-loop structure may have a completely different function from a 100-nt long double helix in living cells. Yet, both structures could share an identical graphical representation under this approach, potentially oversimplifying key structural distinctions.

B. It is commendable that the authors categorized graph topologies into three groups: existing, RNA-Like, and non-RNA-Like, based partly on prior structural knowledge. However, RNA structures exhibit high dynamic variability. Less frequently observed conformations may be misclassified as non-RNA-Like due to overfitting to a limited existing dataset. Additionally, current RNA secondary and tertiary structure prediction tools have inherent technical limitations, which may further affect the accuracy of the graphical representation. Clarifying these limitations in the manuscript is important.

2. Distinction Between ‘RNA-Like’ and ‘Non-RNA-Like’ Graph Classes

A. While the authors discuss that biological RNA topologies are more likely to contain distinguishing subgraphs, a more comprehensive explanation and comparison between RNA-Like and non-RNA-Like graph classes is necessary. This should incorporate insights from clustering, persistent subgraph generation (PSG), or other data mining approaches.

B. If persistent Betti numbers (Betti 0 and Betti 1) can be mapped back to specific biological or structural scenarios, the interpretation will be more intuitive. For instance, if one category consistently retains more connected components, this should be explicitly analyzed and explained.

**Have the authors made all data and (if applicable) computational code underlying the findings in their manuscript fully available?**

Reviewer #1: Yes

Reviewer #2: Yes

PLOS authors have the option to publish the peer review history of their article (what does this mean?). If published, this will include your full peer review and any attached files.

Reviewer #1: No

Reviewer #2: No

**Figure resubmission:**
---

## [Decision Letter · Decision Letter 1]

12 Jun 2025

Dear Dr. Wang,

We are pleased to inform you that your manuscript 'How Large is the Universe of RNA-Like Motifs? A Clustering Analysis of RNA Graph Motifs Using Topological Descriptors' has been provisionally accepted for publication in PLOS Computational Biology.

Best regards,

Mingfu Shao, Ph.D.

Academic Editor

PLOS Computational Biology

Ilya Ioshikhes

Section Editor

PLOS Computational Biology

Reviewer's Responses to Questions

**Comments to the Authors:**

Reviewer #1: The authors have addressed all my comments and I believe this substantially improved manuscript is suitable for publication.

Just one more optional comment: I still think it is confusing and uncessary to include the concept of persistence, as it is not used at all in this work. This to me is the same as calling an ODE a special case one-independent-variable PDE throughout a paper that only discusses ODE. The generalization to persistence cases can be mentioned in the discussion section but it is not necessary to use this term throughout the manuscript. That being said, this does not affect the main biochemical message and I respect whichever the authors choose.

Reviewer #2: The authors have addressed my comments.

**Have the authors made all data and (if applicable) computational code underlying the findings in their manuscript fully available?**

Reviewer #1: Yes

Reviewer #2: Yes

PLOS authors have the option to publish the peer review history of their article (what does this mean?). If published, this will include your full peer review and any attached files.

Reviewer #1: No

Reviewer #2: **Yes: **Yiliang Ding

---

## [Editor Report · Acceptance letter]

PCOMPBIOL-D-25-00058R1

How Large is the Universe of RNA-Like Motifs? A Clustering Analysis of RNA Graph Motifs Using Topological Descriptors

Dear Dr Wang,

I am pleased to inform you that your manuscript has been formally accepted for publication in PLOS Computational Biology. Your manuscript is now with our production department and you will be notified of the publication date in due course.

With kind regards,

Anita Estes
